# Diversity of Linear Non-Ribosomal Peptide in Biocontrol Fungi

**DOI:** 10.3390/jof6020061

**Published:** 2020-05-12

**Authors:** Xiaoyan Niu, Narit Thaochan, Qiongbo Hu

**Affiliations:** 1Key Laboratory of Bio-Pesticide Innovation and Application of Guangdong Province, College of Agriculture, South China Agricultural University, Guangzhou 510642, China; ny862548923@163.com; 2Pest Management Biotechnology and Plant Physiology Laboratory, Faculty of Natural Resources, Prince of Songkla University, Hat Yai, Songkhla 90110, Thailand; narit.t@psu.ac.th

**Keywords:** entomopathogenic fungi, mycoparasitic fungi, linear NRPs, diversity

## Abstract

Biocontrol fungi (BFs) play a key role in regulation of pest populations. BFs produce multiple non-ribosomal peptides (NRPs) and other secondary metabolites that interact with pests, plants and microorganisms. NRPs—including linear and cyclic peptides (L-NRPs and C-NRPs)—are small peptides frequently containing special amino acids and other organic acids. They are biosynthesized in fungi through non-ribosomal peptide synthases (NRPSs). Compared with C-NRPs, L-NRPs have simpler structures, with only a linear chain and biosynthesis without cyclization. BFs mainly include entomopathogenic and mycoparasitic fungi, that are used to control insect pests and phytopathogens in fields, respectively. NRPs play an important role of in the interactions of BFs with insects or phytopathogens. On the other hand, the residues of NRPs may contaminate food through BFs activities in the environment. In recent decades, C-NRPs in BFs have been thoroughly reviewed. However, L-NRPs are rarely investigated. In order to better understand the species and potential problems of L-NRPs in BFs, this review lists the L-NRPs from entomopathogenic and mycoparasitic fungi, summarizes their sources, structures, activities and biosynthesis, and details risks and utilization prospects.

## 1. Introduction

Biocontrol fungi (BFs) play an important role in the control of agricultural and forestry pests. BFs include mainly entomopathogenic and mycoparasitic fungi (EFs and MFs). Entomopathogenic fungi are used extensively in agricultural and medical areas. *Beauveria bassiana* and *Metarhizium anisopliae* have been developed as commercial BFs to manage many insect pests worldwide; *Cordyceps* spp. have been used in traditional medicines in Asia for many years [1,2]. Mycoparasitic fungi such as *Trichoderma* spp. have been used to control soil-borne plant diseases at commercial scales [3,4]. BFs produce multiple secondary metabolites to interact with pests, plants and microorganisms for better adapting to their environments.

Secondary metabolites produced by BFs mainly include polyketides (PKs), terpenes and non-ribosomal peptides (NRPs). NRPs are synthesized by multidomain mega-enzymes named nonribosmal peptide synthetases (NRPSs), without ribosomes and messenger RNAs. NRPSs assemble numerous NRPs with large structural and functional diversity, including more than 20 marketed drugs with antibacterial (penicillin, vancomycin), antitumor (bleomycin) and immunosuppressant (cyclosporine) activities [5]. Apart from the 20 protein amino acids, NRPs also contain rare amino acids and other organic acids. The N-terminal of NPRs are often modified by fatty acids, heterocyclic compounds, glycosylated or phosphorylated structures [6]. NRPs are divided into linear (L-NRPs) and cyclic NRPs (C-NRPs). Due to lack of cyclization, L-NRPs have peptide chains composed of multiple amino acids often modified by different fat chains or non-protein amino acids. L-NRPs have antimicrobial, insecticidal, antiviral or anticancer properties. There are numerous studies and reviews on fungal non-ribosomal synthases and cyclic peptides [7,8,9]. However, little attention has been paid to L-NRPs from BFs. As the agricultural and medical importance of BFs, it is necessary to investigate the BFs L-NRPs source species, structure, activity and biosynthesis, as well as their potential risks.

## 2. L-NRPs from Entomopathogenic Fungi

There are more than 1000 species of EFs, mainly belonging to the order Hypocreales. However, only seven L-NRPs have been reported in a few species of the genera, *Cordyceps, Paecilomyces*, *Metarhizium* and *Hirsutella*.

### 2.1. Cicadapeptins

Cicadapeptins are obtained from *Cordyceps heteropoda* ARSEF1880 [10]. There are two analogs of cicadapeptins with a modified N-terminal hydroxyproline (Hyp) acylated by n-decanoic acid and a C-terminal leucine (Leu) modified by 1,2-diamino-4-methylpentane (Figure 1). The two residues of Aib (alpha-aminoisobutyric acid) in the chain lead to the typical helical structure of cicadapeptins. Moreover, the Hyp came out twice in succession results in the structural change from helical to continuously curved one [11]. Cicadapeptins I and II show inhibitory effects on Gram-positive and negative bacteria [11]. 

### 2.2. Hirsutellic Acid A

Hirsutellic acid A (Figure 2) is a tripeptide isolated from *Hirsutella* sp. BCC 1528. It has non-polar amino acids Ile, Leu and N-methyl-phenylalanine (N-Me-Phe) [12]. Hirsutellic acid A is inhibitory against *Plasmodium falciparum* K1 with an IC_50_ of 8 μM [12].

### 2.3. Leucinostatins

Leucinostatins can be found in *Purpureocillium lilacinum* (*Paecilomyces lilacinus*), *Metarhizium marquandii* (*Paecilomyces marquandii*) and *Acremonium* sp. [13,14,15]. About 24 analogs of leucinostatins were isolated and identified (Figure 3; Table 1). Leucinostatins have different biologic activities, such as antibacterial, antifungal, cytotoxic and phytotoxic properties with inhibition of oxidative phosphorylation and ATP synthesis [16]. Leucinostatins have higher activity against nematodes with LD_50_ of 10 mg/L [17]. Leucinostatin A inhibits the growth of prostate cancer cells by hindering insulin-like growth factor-I expression [18]. Leucinostatins also have inhibitory effects on plant parasitic oomycetes (*Phytophthora* spp.) [13].

### 2.4. Efrapeptins

Efrapeptins are mainly produced by *Tolypocladium niveum* (*Tolypocladium inflatum*, *Beauveria nivea*) and *Tolypocladium geodes* [19,20,21] but are also found in *Acremonium* sp. and *Metarhizium anisopliae* [20,22]. Efrapeptins have 10 analogs with molecular weight of 1592–1676 Da [22]. The analogs of efrapeptins contain alpha–azide carboxylic acid (Figure 4, Table 2). Efrapeptins have antifungal, antimalarial, insecticidal and antitumor activities [20,23]. In particular, they can inhibit the ATPase and, disturb the interaction between ATPase and heat shock protein 90 (Hsp90) [24,25,26]. Actually, not only the F1-ATPase, but also the vacuolar ATPase (V-ATPase) in brush border membrane vesicles (BBMV) can be inhibited [27]. Efrapeptin J inhibit gastric cancer cells HT1080 [28].

### 2.5. Peptaibol Compounds

Peptaibols are a special kind of L-NRPs which have been found in a variety of soil fungi. Most of them are found in *Trichoderma* spp. which have been used to control plant disease [29]. To date, more than 500 peptaibols have been identified [30] and among them, 35 types were identified after 2000 [31]. Peptaibols are rich in non-protein amino acid and are often acetylated at the N-terminus and hydroxylated at the C-terminus. Peptaibols contain 5–20 aa residues forming α-helical conformation [32,33]. Moreover, peptaibols form ion channels on the lipid bilayer membrane. Those peptaibols with long sequences (12–20 aa) have the "barrel–stave" ion channel, while, the others with short sequences (5–11 aa) possess the "carpet" ion channel as a dimers with their N-termini connected (Figure 5) [34]. Therefore, peptaibols can break the ion balance of cells leading to functional disorder of cell. They not only have antibacterial, cytotoxic activity, but also are teratogenic to the larvae of some marine organisms [35,36]. In BFs, there are three peptaibols reported.

#### 2.5.1. Culicinins

The entomopathogenic fungus *Culicinomyces clavisporus* LL-12I252 [37,38] produces culicinins. They are decapeptides with four analogs (A–D) with the sketch of Bta-Pro-Ahmod-Aib/Ala-Aib/Ala-Amd-Leu-Aib-Leu-Apa-Apae-OH (Figure 6). Culicinin D can selectively inhibit MDA 468 breast cancer cells [38].

#### 2.5.2. LP237

The entomopathogenic fungus *Tolypocladium geodes* (*Beauveria geodes*) produces LP237 with three analogs (Table 3) [39,40,41]. The highly helical structure of LP237 and the amphiphilic side chain of amino acids form a “barrel–stave” ion channel on the membrane, the Gln_6_, Gln_7_ and Gln_10_ in the peptide are on the same polar surface of the helix, forming a cavity of the ion channel, resulting in the membrane permeability activity of the peptide [42]. LP237 F8 is cytotoxic to P388D1 mouse leukemia cells and human tumor cells, such as lung cancer A549, ovarian cancer OVCAR3, colon cancer SW620 and breast cancer MCF7. It has synergistic actions with other anticancer peptides [42].

#### 2.5.3. Metanicins

Metanicins are produced by strain CBS 597.80 of *Metarhizium*. There are four analogs with 20 aa residues and are similar to the peptaibols (longibrachin, trichobrachins, trichokonins) from *Trichoderma* (Table 4). Metanicins inhibit bacteria with the activity order of *Micrococcus lutes* > *Enterococcus faecalis* > *Staphylococcus aureus* > *Bacillus subtilis* [43].

## 3. L-NRPs from Mycoparasitic Fungi

There were 14 L-NRPs found in mycoparasitic fungi mainly in the genus of *Trichoderma* and *Penicillium*. Apart from δ-(L-α-aminoadipyl)-L-cysteinyl-D-valine (ACV), the other 13 L-NRPs all belong to peptaibols.

### 3.1. ACV

ACV is a tripeptide formed by condensation of L-aminoadipic acid, L-cysteine and L-valine (Figure 7). It is a synthetic precursor of the antibiotics penicillin and cephalosporins [44]. ACV was isolated from *Penicillium chrysogenum*, *Cephalosporins acremonium* and *Aspergillus nidulans*. Interestingly, *Penicillium chrysogenum* is the most important ACV producer, it not only increases plant resistance to pathogens [45], but also has insecticidal activity to *Bactrocera oleae* [46]. ACV is synthesized by ACV synthase (ACVS) which was from fungi and bacteria [47,48,49].

### 3.2. Harzianins

Harzianins are named because first, identified in extract of *Trichoderma harzianum*. *T. harzianum* not only has a good inhibitory effect on plant pathogens, but also be used to control mosquito pests [50]. To date, there are up to 15 harzianin analogs (Table 5). The HC type contains three kinks formed by the Aib-Pro motifs. The structures are 3_10_-helices, which are embedded in the lipid layer to form a voltage-gated ion channel of the "barrel–stave" type, which increases the hydrophobicity and permeability of the lipid bilayer [51,52]. The PCU4 type are similar to HC, but with shorter chain [53]. Compared to HC type, the HB I is missing an Aib-Pro-Ala [54]. There are two Aib-Pro motifs in the HK VI, also in the 3_10_-helices conformation [55]. The HA V only contains an Aib-Pro, forming the center hinge of Pro’s α-helix structure [56].

### 3.3. Trichorzins

Trichorzins are 18 aa peptaibols with up to 10 analogs found in *T. harzianum* and *T. virens* (Table 6). Trichorzin PAs with six analogs found in *T. harzianum* show the higher activity against mycoplasma and spiroplasma [57,58]. Three TVB analogs were isolated from *T. virens* [59]. Trichorzins PAs have a polar C-terminus of tryptophan (Trpol) with affinity to the hydrophilic head of the phospholipid molecule in bilayer membrane, which is important for construction of a voltage-gated ion channel of these “barrel–stave” peptaibols [60].

### 3.4. Longibrachins

*Trichoderma longbrachitum* has strong inhibition against the soil-borne phytopathogens, *Rhizoctonia solani*, *Sclerotium rolfsii* and *Pythium aphanidermatum* [61]. In addition, it also has insecticidal and nematocidal activity, respectively against insect *Leucinodes orbonalis* and *Heterodera avenae* [62,63].

*T. longibrachiatum* produces longibrachin (LG), which is a peptaibol with 20 aa residues. Six LG analogs were found. The A series LGs with four analogs (LG A Ⅰ–Ⅳ ) have the neutral Gln at 18th aa residues, while the B series (LG B Ⅱ–Ⅲ) are replaced with acidic Glu (Table 7) [64,65]. The negatively charged side chain Glu of LG B increases the oligomerization level of the ion channel and improves the transportation of substances [66]. LGs result in deformities of *Crassostrea gigas* larvae and may be neurotoxic to *Calliphora vomitoria* with an ED_50_ of 270 mg/kg [64,67]. They are also toxic effect on KB cells (human oral epidermoid cancer cells) [64]. LGs show antibacterial activity against mycoplasma and Gram-positive bacteria. LG AIV also shows a weak inhibition of human pathogenic fungus *Aspergillus fumigatus* [64,65].

### 3.5. Trichobrachins

Trichobrachins are purified from *T. longibrachiatum* and *T. parceramosum* [68]. There are 11 analogs (Table 8) [68,69]. Trichobrachins inhibit *Bacillus subtilis* [68].

### 3.6. Trichogins

Trichogin GA IV is purified from *T. longibrachiatum* strain M3431. It has a sequence of nOct-Aib-Gly-Leu-Aib-Gly-Gly-Leu-Aib-Gly-Ile- Leuol (Figure 8) [70,71]. Trichogin has a N-terminus of n-octanoic acid and contains four hydrophobic glycine, which lead to a hybrid structure with the 3_10_-helix and the α-helix of trichogen [72,73]. Trichogin has low hemolytic activity and obvious antibacterial activity against methicillin-resistant *Staphylococcus aureus* [72,74]. Trichogin is also toxic to several human tumor cells (mononuclear HL60 in myelogenous leukemia, HeLa in ovarian sarcoma, A431 in epidermoid cancer and A549 in lung cancer) [73,75].

### 3.7. Trilongins

Trilongins have 13 analogs with 11 or 20aa residues and are mainly found in *T. longibrachiatum* and *T. atroviride* (Table 9) [76,77]. The trilongin A series have 11aa residues with average molecular weight of 1175 Da, while the trilongin B and C series have 20aa residues with 1936–1965 Da. Trilongins are toxic to mammals. They destroy the mitochondria of boar sperm cells, remarkably, the mixtures of long and short sequences trilongins are more toxic [77]. Trilongins form voltage-gated K^+^/Na^+^ ion channels, moreover, the combinations of A type with B/C type than the single type have synergistic effect to keep the ion channel open longer [77].

### 3.8. Trichokonins

*Trichoderma koningii* and *Trichoderma pseudokoningii* are common mycoparasitic fungi used to control the fungal phytopathogens of vegetables and fruit trees diseases [78,79]. They can produce trichokonins, which are also found in *T. longibrachiatum*. Trichokonins have four analogs containing large amounts of Aib and forming a helical conformation with a kink (Pro) (Table 10) [64,80,81]. Trichokonins have a broad antibacterial activity against Gram-positive bacteria, especially to *Staphylococcus aureus*, even multidrug-resistant *Staphylococcus aureus*. However, trichokonins have no significant effect on Gram-negative bacteria. In addition, trichokonins are highly toxic to hepatocellular carcinoma cells HepG2, lung cancer cells A549 and gastric cancer cells BGC823, leading then to apoptosis [82]. Tichokonin VI is an L-type Ca^2+^ agonist on cardiac biofilms [83]. In addition, trichokonins also promote antiviral activity of tobacco by inducing the defense response and systemic resistance of tobacco to tobacco mosaic virus (TMV) infection [80].

### 3.9. Alamethicins

*Trichoderma viride* (NRRL 3199), a BF widely distributed in nature and used to control soil-borne plant diseases [84,85], produces alamethicins with two analogs, B30 and B50 (Table 11) [86]. Each analogs has many derivatives with the absences of the N-terminal six residues or the C-terminal phenylalaninol (pheol) or the substitution of Ala of 6th residue with Aib or Gln of 7th and 19th residues with Glu. Alamethicins rich in Aib and have two Pro near the N-terminal and C-terminal, the N-terminal of the molecule forms a stable α-helix and the C-terminal exhibits a variable hydrogen bonding pattern [87]. Alamethicin is often used as a model ion channel for passive diffusion of voltage-gated cation ions [88,89].

### 3.10. Trichotoxins

Trichotoxins with 18 residues were purified from *T. viride* and *T. asperellum*. These compounds are divided into trichotoxin A-40 and A-50 analogs with multiple derivatives (Table 12) [90,91]. The differences among the derivatives are the substitution of Aib/Ala, Gln/Glu and C-terminal Aib/D-Iva and only A-50 contains neutral Gln. Trichotoxins have α-helical conformation and form ion channels similar to alamethicin [90]. They cause hemolysis in human erythrocytes at micromolar concentrations and possess cytotoxic activity against GH(4)C(1) rat pituitary and Neuro-2a mouse neuroblastoma cells [92,93].

### 3.11. Suzukacillins

Suzukacillins (SZs) are also purified from *T. viride* [94]. There are two families, suzukacillin A (SZ-A) and suzukacillin B(SZ-B) (Table 13) [94,95]. Suzukacillins exhibit membrane modifying and lysing properties similar to those of alamethicin. Suzukacillins inhibit the growth of *Bacillus subtilis*, *Xanthomonas oryzae*, *Sarcina lutea*, *Staphylococcus aureus*, *Mycobacterium phlei*, *Streptococcus pyogenes*, *Corynebacterium diphtheriae*, *Aspergillus niger*, *Trichophyton fumigatus* and *Saccharomyces sake* at concentrations of 10 to l00 μg/L. Suzukacillins also have hemolytic properties against human erythrocytes [93].

### 3.12. Hypomurocins

Hypomurocins have 13 analogs (A and B series) (Table 14) purified from *Hypocrea muroiana* which is a BF not only inhibiting various plant diseases, but also promoting plant growth [96], excepting hypomurocin B that is found in *Trichoderma harzianum* [59,97]. Hypomurocin A has mixed helical conformation containing α- and 3_10_-helices, as well as types I and III β-turn structures to link the helical [98,99,100]. Hypomurocin B consists of 18 amino acid residues to form the 3_10_-helical structure rather than by α-helical structure [101]. Hypomurocins inhibit *Bacillus subtilis* and causes hemolysis of rat erythrocytes, moreover, the activity of hypomurocin B is greater than that of hypomurocin A [96].

### 3.13. Atroviridins and Neoatroviridins

Atroviridins are 20-residue peptaibols with three analogs (Table 15) [102,103]. Neoatroviridins are 18-residue peptaibols with four analogs (Table 16) [104]. They are all purified from *T. atroviride* (F80317), which is a BFs with antifungal activity [104,105]. Atroviridins and neoatroviridins have strong antifungal activity against some plant pathogenic fungi, such as *Curvavularia inaqualis*, *Collectotrichum dematium* and *Fusarium oxysporum*, as well as moderate activity against *Verticillium dahliae*, *Aspergillus niger* and *Cladosporium* sp. They can inhibit *Bacillus subtilis* and *Staphylococcus aureus*. However, they show no activity against Gram-negative bacteria and yeast [102]. Moreover, they have significant cytotoxicity against human cancer cell lines, such as prostate (PC-3), melanoma (UACC62) and leukemia (K562) with IC_50_ values of 2–4 μg/mL [102].

### 3.14. Peptaivirin

Peptaivirins are special peptaibols purified from *Trichoderma* spp. (KGT142). Peptaivirins have two analogs (Table 17), peptaivirins A and B, which show strong antiviral effects on TMV infection [106]. Peptaivirins are rich in Aib and have an N-terminus of acetylated phenylalanine.

## 4. Biosynthesis of L-NRPs

Non-ribosomal peptides are synthesized by non-ribosomal peptide synthase (NRPS) with multiple modules in some of the largest enzymes found in nature. The modules consist of different domains with specific catalytic activities. The core domains of NRPS include adenylation domain (A domain, recognizing and adenylating the initiation molecule), thiolation domain (T domain, also known as peptidyl carrier protein domain (PCP)) and condensation domain (C domain, catalyzing the corresponding monomers to bind to the new peptide) [8]. In addition to the infrastructure domains (A, T and C domains), NRPSs probably have epimerization domain (E domain), N-methylation domain (M domain) and others to modify the peptide [104]. Finally, there is a thioesterase domain (TE) in bacteria NRPSs or a similar condensation domain (CT domain) in fungal NRPSs to hydrolyze or cyclize the end of the target polypeptide [107,108].

NRPSs are divided into three categories, namely linear, iterative and nonlinear NRPS (Figure 9). The linear NRPSs take C-A-T as the extension module, and the assembly results in the production of linear NRPs or cyclic NRPs. The iterative NRPS has multiple same modules and results in final product of oligopeptides or cyclic NRPs with the multiple residues of the same amino acids. The nonlinear NRPS has other modules (X) and the order of C-A-T is not necessary. It deviates completely from the standard domain organization leading to unexpected products [109]. L-NRPs are mainly synthesized by linear NRPs [109]. The number of modules determines the length of the peptide. Compared with cyclic NRPs, the main difference is whether initiation substrates in A domain have free hydroxyl and amidogen. After hydrolysis in the TE domain, internal esterification or lactam hydrolysis will occur [6]. The biosynthesis process of L-NRPs in biocontrol fungi is complex and few researches have been published, so we take ACV as an example to illustrate the biosynthesis of L-NRPs.

The *PcbAB* of ACV synthetase (ACVS) was cloned from *P. chrysogenum*, it measures 11,500 bp with the open reading frame (ORF) being 11,376 bp and coding for a protein of 3791 aa. The genes, *PcbAB*, *PcbC* (encoding cyclase) and *PcbDE* (encoding penicillin acetyltransferase) form a cluster in the 17 Kb DNA region to drive penicillin biosynthesis [110]. The ACVS genes of *Cephalosporins acremonium* and *Aspergillus nidulans* are similar to those of *P. chrysogenum*, with more than 60% similarity [111,112]. ACVS contains ten domains, three modules (M1, M2 and M3), in which M3 has the special domains E and TE domains to conduct epimerization of L-valine and the hydrolysis of ACV (Figure 10) [113]. During biosynthesis, A domain in modules M1 chooses the suitable substrate L-α-aminoadipic acid to activate and form an aminoacyl-AMP. Then, it combines with hydrosulfonyl of T domain to form aminoacyl-S-carrier complex and transferred to modules M2 and form cysteinyl-aminoacyl-S-carrier complex by combining with the activated cysteinyl-S-carrier. Then, it is transferred to M3 and condensated with the activated valinyl-S-carrier into valinyl-cysteinyl-aminoacyl-S-carrier complex. Finally, through intramolecular nucleophilic attacks in TE domain, the L-NRPs δ-(L-α-amino hexanedioyl)-L-cysteinyl-D-valine (ACV) is produced (Figure 10).

## 5. Discussion

Although only 22 classes of L-NRPs are found in BFs to date, BFs absolutely have abundant diversity of L-NRPs. First, the BFs L-NRPs have diverse molecular structures, i.e., each class has multiple analogs and numerous derivatives with different configurations and conformations. Second, the BFs L-NRPs have multiple functions because each has diverse bioactivities among of antifungi, antibacteria, antiviruses, insecticides, acaricides, nematicides, herbicides or anticancers. Finally, the BFs L-NRPs have diverse distribution, i.e., one species of BFs has more than one class of L-NRPs, on the other hand, a same L-NRP can exist in different BFs species. For example, *Trichoderma harzianum* at least has two L-NRPs, harzianins and hypomurocin B, while *Trchoderma longibrachiatum* produces trichobrachins, trichogins, trilongins and trichokonins. Furthermore, efrapeptins are produced by *Tolypocladium niveum*, *Tolypocladium geodes*, *Acremonium* sp. and *Metarhizium anisopliae*, while trichokonins can be found in *Trichoderma koningii*, *Trichoderma pseudokoningii* and *T. longibrachiatum.*

Interestingly, more L-NRPs have been found in mycoparasitic fungi than in entomopathogenic fungi, especially in the common entomopathogens, such as *Beauveria*, *Metarhizium* and *Isaria.* The main reason may be related to the L-NRPs characteristics of easy hydrolysis [114]. Insects have many proteases especially in their midguts, so if entomopathogens secrete L-NRPs into insect’s body, they will be hydrolyzed soon. However, C-NRPs are difficult to degradation in insects. On the contrary, mycoparasitic fungi usually live in soil and interact with phytopathogens or other microorganisms—in an environment with less proteases. Therefore, the L-NRPs secreted by mycoparaites may persist for a longer time, which has beneficial influences on surrounding microorganisms. The diversity of NRPs is ensured by NRPS through different organizations of domains and modules. The A domains with various structures can select different substrate amino or fatty acids to provide the diverse composition of peptide chain. Undoubtedly, to adapt environments, BFs must take the least costs to obtain the best NRPS genes. Such, the co-evolution of BFs and these target lives leads to less L-NRPs existing in entomopathogenic fungi than in mycoparasitic fungi.

NRPs as drug resources attract much attention of researchers. BFs L-NRPs have the potential as pesticides and medicines as well. For example, bleomycin has been used to treat cancers [5]. ACV as a precursor compound of penicillin has been concerned for long times [52]. Although NRPs are currently not used in agricultural area, the further studies are valuable. However, the more important is the risks of L-NRPs in BFs. As many L-NRPs are toxic, they can hazard human health and non-target beings once they enter the food chain in the process of agricultural application. Although NRPs produced by BFs have little probability to enter food chain [2,9,115], caution must be exercised. It is necessary that adequate risk assessments are conducted before using BFs.

In conclusion, there are 22 classes L-NRPs found in BFs currently. They have abundant diversity including various structures, functions and distributions. The NRPSs through different compositions of domains and modules accomplish biosynthesis of deferent L-NRPs. Mycoparasitic fungi than entomopathogenic fungi produce more L-NRPs, it is maybe because the co-evolutions of fungi with their hosts lead to NRPSs in these two fungi. BFs L-NRPs have the potential as pesticides and medicines. However, the risks of L-NRPs contaminating foods and environment need be paid more attentions.

## Figures and Tables

**Figure 1 jof-06-00061-f001:**
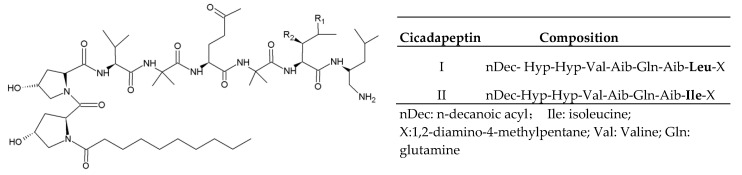
Basic structure and analogs of cicadapeptins

**Figure 2 jof-06-00061-f002:**
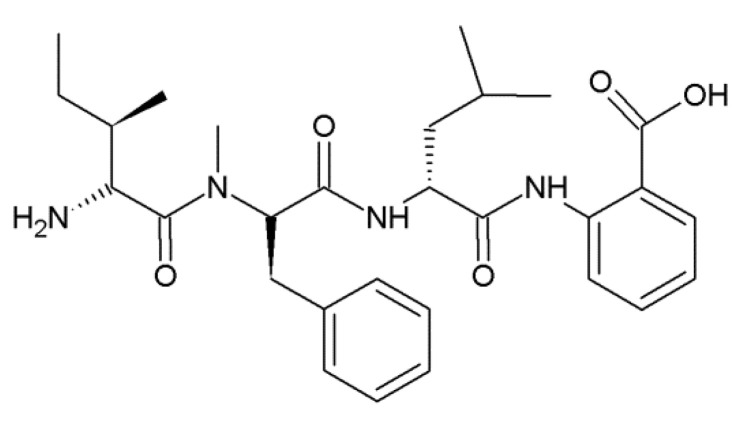
Structural formula of hirsutellic acid A.

**Figure 3 jof-06-00061-f003:**
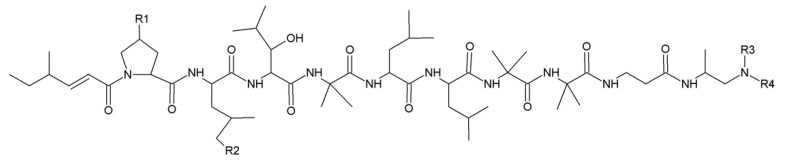
Basic structure of leucinostatins.

**Figure 4 jof-06-00061-f004:**
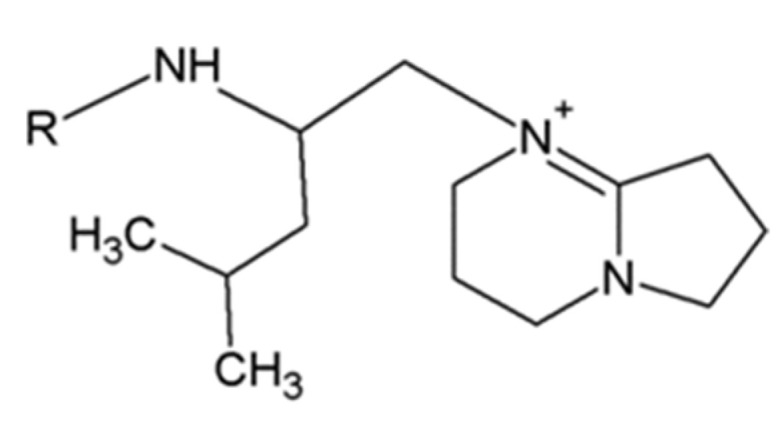
Basic structure and analogs of efrapeptins.

**Figure 5 jof-06-00061-f005:**
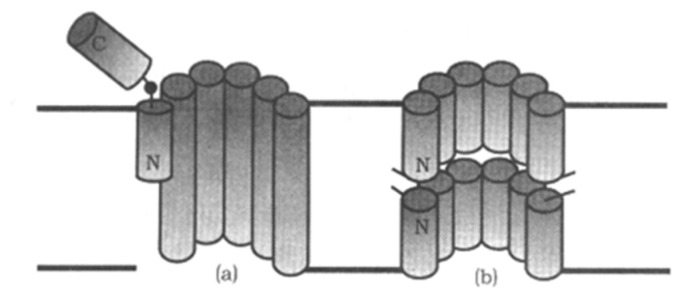
Peptaibols act as ion channels in membrane [34]. (**a**) “barrel–stave” ion channel model; (**b**) “carpet” ion channel model.

**Figure 6 jof-06-00061-f006:**
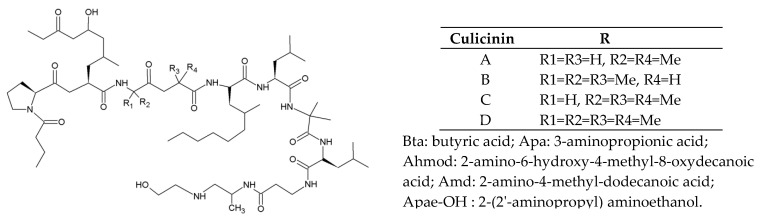
Basic structure and analogs of culicinins.

**Figure 7 jof-06-00061-f007:**
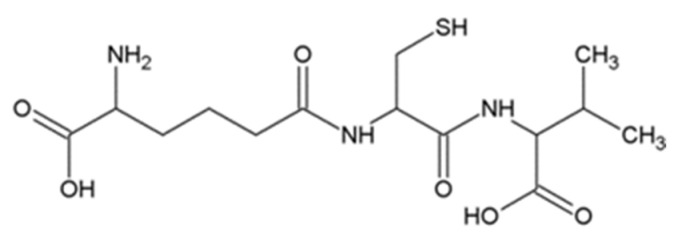
Structural formula of δ-(L-α-aminoadipyl)-L-cysteinyl-D-valine.

**Figure 8 jof-06-00061-f008:**
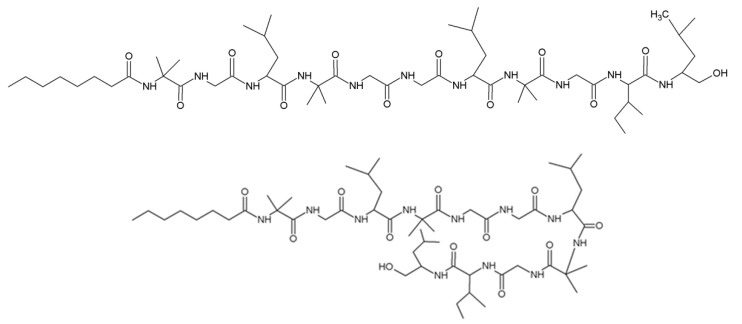
Structure of trichogin GA IV.

**Figure 9 jof-06-00061-f009:**
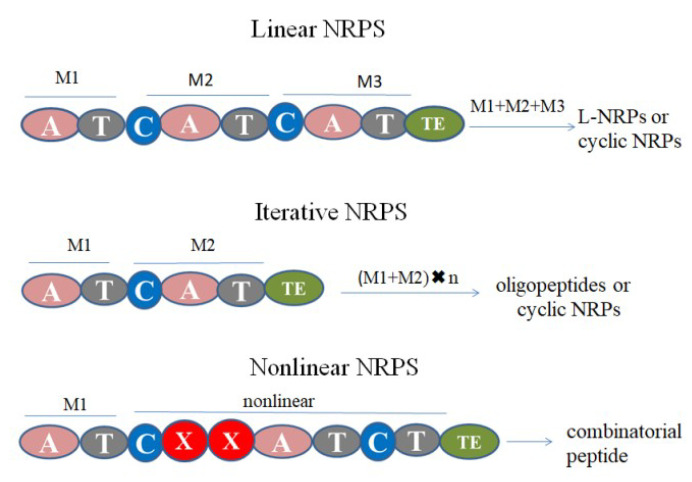
Synthesis of non-ribosomal peptide synthases (NRPSs) types. (M1 module is the initial of synthesis, M2 and M3 are the extension; X in nonlinear NRPS represents other domains).

**Figure 10 jof-06-00061-f010:**
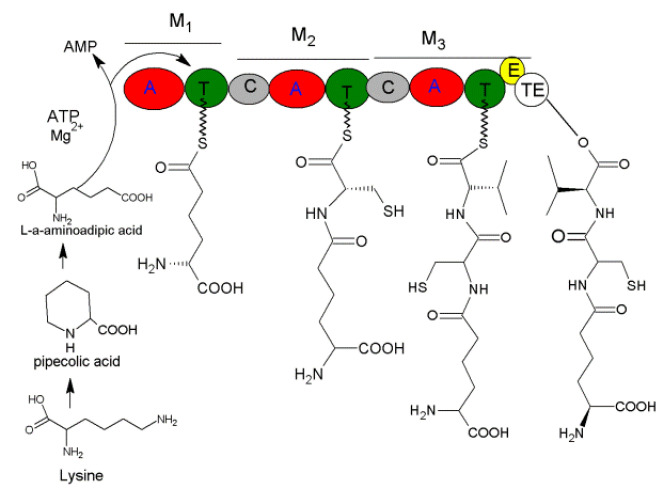
Biosynthesis of ACV.

**Table 1 jof-06-00061-t001:** Analogs of leucinostatins.

Leucinostatin	Strain	R_1_	R_2_	R_3_	R_4_	Ref.
C	*P. lilacinus* CG-189	-CH_3_	-H	-H	-H	[13]
T	*P. lilacinus* CG-189	-H	-H	-H	-CH_3_	[13]
F	*P. lilacinus* CG-189	-CH_3_	-H	-H	-CH_3_	[13]
D	*P. lilacinus* CG-189	-CH_3_	-H	-CH_3_	-CH_3_	[13]
N	*P. lilacinus* CG-189	-CH_3_	-OH	-H	-CH_3_	[13]
H *	*P. lilacinu* CG-189 *P. marquandii*	-CH_3_	-H	\	\	[13,14]
B_2_	*P. lilacinus* CG-189	-CH_3_	-CH=CHCOCH_2_CH_3_	-H	-CH_3_	[13]
V	*P. lilacinus* CG-189	-CH_3_	-CH_2_CH_2_COCH_2_CH_3_	-H	-CH_3_	[13]
L	*P. lilacinus* CG-189	-H	-CH(OH)CH_2_COCH_2_CH_3_	-H	-CH_3_	[13]
A_2_	*P. lilacinus* CG-189	-CH_3_	-CH=CHCOCH_2_CH_3_	-CH_3_	-CH_3_	[13]
R	*P. lilacinus* CG-189	-CH_3_	-CH_2_CH_2_COCH_2_CH_3_	-CH_3_	-CH_3_	[13]
B	*P. lilacinus* CG-189	-CH_3_	-CH(OH)CH_2_COCH_2_CH_3_	-H	-CH_3_	[13]
S	*P. lilacinus* CG-189	-CH_3_	-CH_2_CH_2_(OH)CHCH_2_CH_3_	-CH_3_	-CH_3_	[13]
A	*P. lilacinus* CG-189	-CH_3_	-CH(OH)CH_2_COCH_2_CH_3_	-CH_3_	-CH_3_	[13]
U	*P. lilacinus* CG-189	-CH_3_	-CH(OCH_3_)CH_2_COCH_2_CH_3_	-H	-CH_3_	[13]
K	*P. lilacinus* CG-189, *P. marquandii*	-CH_3_	-CH(OH)CH_2_COCH_2_CH_3_	\	\	[13,14]
W	*P. lilacinus* CG-189	-CH_3_	-(C_10_H_15_O_3_)	-H	-CH_3_	[13]
Q	*P. lilacinus* CG-189	-CH_3_	-(C_11_H_17_O_3_)	-H	-CH_3_	[13]
O	*P. lilacinus* CG-189	-CH_3_	-(C_11_H_17_O_3_)	-CH_3_	-CH_3_	[13]
Ⅰ	*P. lilacinus* CG-189	-H	-CH_3_	-CH_3_	-CH_3_	[13]
Ⅱ	*P. lilacinus* CG-189	-H	-H	-H	-CH_3_	[13]
Ⅲ	*P. lilacinus* CG-189	-H	-CH_2_CH_2_(OH)CHCH_2_CH_3_	-CH_3_	-CH_3_	[13]
Ⅳ	*P. lilacinus* CG-189	-CH_3_	-CH_2_CH_2_(OH)CHCH_2_CH_3_	-H	-CH_3_	[13]
V	*P. lilacinus* CG-189	-H	-CH(OH)CH_2_COCH_2_CH_3_	-CH_3_	-CH_3_	[13]

* The C- terminal -N(R_3_)R_4_ of leucinostatin H is altered with -CH_2_CH_2_ONH_2_.

**Table 2 jof-06-00061-t002:** The analogs of efrapeptins.

Efrapeptin	Strain	R	Ref.
A	*T. niveum* ARSEF NO.616	Ac-Aib-Gly-Leu-Iva-	[20]
B	*T. niveum* ARSEF NO.616	Ac-Leu-Iva-	[20]
C	*T. niveum* ARSEF NO.616, *T. geodes* ARSEF 2684	Ac-Pip-Aib-Pip-Aib-Aib-Leu-βAla-Gly-Aib-Aib-Pip-Aib-Gly-Leu-Aib-	[19,20]
D	*T. niveum* ARSEF 616, *T. geodes* ARSEF 2684, *M. anisopliae* ME1	Ac-Pip-Aib-Pip-Aib-Aib-Leu-βAla-Gly-Aib-Aib-Pip-Aib-Gly-Leu-Iva-	[19,20]
E	*T. niveum* ARSEF 616, *T. geodes* ARSEF 2684	Ac-Pip-Aib-Pip-Iva-Aib-Leu-βAla-Gly-Aib-Aib-Pip-Aib-Gly-Leu-Iva-	[19,20]
E_α_	*Acremonium* sp. 021172c	Ac-Pip-Aib-Pip-Iva-Aib-Leu-βAla-Gly-Aib-Aib-Pip-Aib-Ala-Leu-Aib-	[22]
F	*T. niveum* ARSEF 616, *T. geodes* ARSEF 2684, *M. anisopliae* ME1	Ac-Pip-Aib-Pip-Aib-Aib-Leu-βAla-Gly-Aib-Aib-Pip-Aib-Ala-Leu-Iva-	[19,20]
G	*T. niveum* ARSEF 616, *T. geodes* ARSEF 2684	Ac-Pip-Aib-Pip-Iva-Aib-Leu-βAla-Gly-Aib-Aib-Pip-Aib-Ala-Leu-Iva-	[19,20]
H	*T. geodes* ARSEF 2684, *Acremonium* sp. 021172c	Ac-Pip-Aib-Pip-Iva-Iva-Leu-βAla-Gly-Aib-Aib-Pip-Aib-Ala-Leu-Iva-	[19,22]
J	*Tolypocladium* sp. AMB18	Ac-Pip-Aib-Pip-Aib-Aib-Leu-βAla-Gly-Aib-Aib-Pip-Aib-Ala-Leu-Aib-	[28]

Ac: acetyl; Pip: 4-aminopiperidine-4-carboxyl; Iva: S-isovaline; βAla: β-Alanine; Gly: glycine.

**Table 3 jof-06-00061-t003:** Analogs of LP237.

LP237	Composition
F8	nOca-Aib-Pro-Phe-Aib-Gln-Gln-Aib-Et-Nva-Gln-Ala-Leuol
F5	nOca-Aib-Pro-Tyr-Aib-Gln-Gln-Aib-Et-Nva-Gln-Ala-Leuol
F7	nDec-Aib-Pro-Phe-Aib-Gln-Gln-Aib-Aib-Gln-Ala-Leuol

Leuol: Leucinol; Et-Nva: 2-amino-3-ethylpentanoic acid; Tyr: Tyrosine.

**Table 4 jof-06-00061-t004:** Analogs of metanicins.

Metanicins	Composition
A	Ac-Aib-Ala-Aib-Ala-Aib-Ala-Gln-Aib-Val-Aib-Gly-Leu-Aib-Pro-Val-Aib-Aib-Gln-Gln-Pheol
B	Ac-Aib-Ala-Aib-Ala-Aib-Ala-Gln-Aib-Val-Aib-Gly-Leu-Aib-Pro-Val-Aib-D-Iva-Gln-Gln- Pheol
C	Ac-Aib-Ala-Aib-Ala-Aib-Aib-Gln-Aib-Val-Aib-Gly-Leu-Aib-Pro-Val-Aib-Aib-Gln-Gln-Pheol
D	Ac-Aib-Ala-Aib-Ala-Aib-Aib-Gln-Aib-Val-Aib-Gly-Leu-Aib-Pro-Val-Aib-D-Iva-Gln-Gln-Pheol

Pheol: Phenylalaninol.

**Table 5 jof-06-00061-t005:** Analogs of harzianins.

Harzianin	Residue	Analogs	Strain	Composition	Ref.
HC	14	11	*T. harzianum* M-903603	Ac-Aib-Gln/Asn-Leu-Aib-Pro-Ala/Ser-Ile/Val-Aib-Pro-Iva/Aib-Leu-Aib-Pro-Leuol	[51]
HB I	11	1	*T. harzianum* M-903603	Ac-Aib-Asn-Leu-Ile-Aib-Pro-Iva-Leu-Aib-Pro-Leuol	[54]
HK VI	11	1	*Trichoderma* sp.	Ac-Aib-Asn-Ile-Ile-Aib-Pro-Leu-Leu-Aib-Pro-Leuol	[55]
HA V	18	1	*T. harzianum* M-903603	Ac-Aib-Gly-Ala-Aib-Iva-Gln-Aib-Val-Aib-Gly-Leu-Aib-Pro-Leu-Aib-Iva-Gln-Leuol	[56]
PCU4	14	1	*T. harzianum*	Ac-Aib-Asn-Leu-Aib-Pro-Ser-Ile-Aib-Pro-Aib-Leu-Aib-Pro-Valinol	[53]

**Table 6 jof-06-00061-t006:** Analogs of trichorzins.

Trichorzin	Strain	Composition	Ref.
PA II	*T. harzianum* M-902608	Ac-Aib-Ser-Ala-Aib-Iva-Gln-Aib-Val-Aib-Gly-Leu-Aib-Pro-Leu-Aib-Aib-Gln-Trpol	[57]
PA IV	*T. harzianum* M-902608	Ac-Aib-Ser-Ala-Aib-Iva-Gln-Iva-Val-Aib-Gly-Leu-Aib-Pro-Leu-Aib-Aib-Gln-Trpol	[57]
PA V	*T. harzianum* M-902608	Ac-Aib-Ser-Ala-Iva-Iva-Gln-Aib-Val-Aib-Gly-Leu-Aib-Pro-Leu-Aib-Aib-Gln-Trpol	[57]
PA VI	*T. harzianum* M-902608	Ac-Aib-Ser-Ala-Aib-Iva-Gln-Aib-Val-Aib-Gly-Leu-Aib-Pro-Leu-Aib-Aib-Gln-Pheol	[57]
PA VIII	*T. Harzianum* M-902608)	Ac-Aib-Ser-Ala-Aib-Iva-Gln-Iva-Val-Aib-Gly-Leu-Aib-Pro-Leu-Aib-Aib-Gln-Pheol	[57]
PA IX	*T. harzianum* M-902608	Ac-Aib-Ser-Ala-Iva-Iva-Gln-Aib-Val-Aib-Gly-Leu-Aib-Pro-Leu-Aib-Aib-Gln-Pheol	[57]
PAU4	*T. harzianum* M-902608	Ac-Aib-Ser-Ala-Aib-Aib-Gln-Aib-Val-Aib-Gly-Leu-Aib-Pro-Leu-Aib-Aib-Gln-Trpol	[57]
TVB I	*T. virens* TV29–8	Ac-Aib-Gly-Ala-Val-Aib-Gln-Aib-Ala-Aib-Ser-Leu-Aib-Pro-Leu-Aib-Aib-Gln-Valol	[59]
TVB II	*T. virens* TV29–8	Ac-Aib-Gly-Ala-Leu-Aib-Gln-Aib-Ala-Aib-Ser-Leu-Aib-Pro-Leu-Aib-Aib-Gln-Valol	[59]
TVB IV	*T. virens* TV29–8	Ac-Aib-Gly-Ala-Leu-Aib-Gln-Iva-Ala-Aib-Ser-Leu-Aib-Pro-Leu-Aib-Aib-Gln-Valol	[59]

Ser: serine.

**Table 7 jof-06-00061-t007:** Analogs of longibrachins.

Longibrachin	Strain	Composition	Ref.
LG A I	*T. longbrachitum* MMS151	Ac-Aib-Ala-Aib-Ala-Aib-Ala-Gln-Aib-Val-Aib-Gly-Leu-Aib-Pro-Val-Aib-Aib-Gln_18_-Gln-Pheol	[64]
LG A Ⅱ	*T. longbrachitum* MMS151	Ac-Aib-Ala-Aib-Ala-Aib-Ala-Gln-Aib-Val-Aib-Gly-Leu-Aib-Pro-Val-Aib-Iva-Gln_18_-Gln-Pheol	[64]
LG A Ⅲ	*T. longbrachitum* MMS151	Ac-Aib-Ala-Aib-Ala-Aib-Aib-Gln-Aib-Val-Aib-Gly-Leu-Aib-Pro-Val-Aib-Aib-Gln_18_-Gln-Pheol	[64]
LG A Ⅳ	*T. longbrachitum* MMS151	Ac-Aib-Ala-Aib-Ala-Aib-Aib-Gln-Aib-Val-Aib-Gly-Leu-Aib-Pro-Val-Aib-Iva-Gln_18_-Gln-Pheol	[64]
LG B Ⅱ	*T. longbrachitum* LCP-853431	Ac-Aib-Ala-Aib-Ala-Aib-Ala-Gln-Aib-Val-Aib-Gly-Leu-Aib-Pro-Val-Aib-Aib-Glu_18_-Gln-Pheol	[65]
LG B Ⅲ	*T. longbrachitum* LCP-853431	Ac-Aib-Ala-Aib-Ala-Aib-Ala-Gln-Aib-Val-Aib-Gly-Leu-Aib-Pro-Val-Aib-Iva-Glu_18_-Gln-Pheol	[65]

**Table 8 jof-06-00061-t008:** Analogs of trichobrachins.

Trichobrachin	Strain	Composition	Ref.
A I	*T. longibrachiatum*	Ac-Aib-Asn-Leu-Leu-Aib-Pro-Leu-Aib-Aib-Pro-Leuol	[69]
A II	*T. longibrachiatum*	Ac-Aib-Asn-Leu-Leu-Aib-Pro-Val-Leu-Aib-Pro-Valol	[69]
A III	*T. longibrachiatum*	Ac-Aib-Asn-Val-Leu-Aib-Pro-Leu-Leu-Aib-Pro-Valol	[69]
A IV	*T. longibrachiatum*	Ac-Aib-Asn-Leu-Val-Aib-Pro-Leu-Leu-Aib-Pro-Valol	[69]
B I	*T. longibrachiatum*	Ac-Aib-Asn-Leu-Leu-Aib-Pro-Val-Aib-Val-Pro-Leuol	[69]
B II	*T. longibrachiatum*	Ac-Aib-Asn-Val-Leu-Aib-Pro-Leu-Aib-Val-Pro-Leuol	[69]
B III	*T. longibrachiatum*	Ac-Aib-Asn-Leu-Val-Aib-Pro-Leu-Aib-Val-Pro-Leuol	[69]
B IV	*T. longibrachiatum*	Ac-Aib-Asn-Leu-Leu-Aib-Pro-Leu-Aib-Val-Pro-Valol	[69]
I	*T. parceramosum* CBS 936.69	Ac-Aib-Ala-Ala/Aib-Ala-Aib-Ala/Aib-Gln-Aib-Vxx-Aib-Gly-Leu-Aib-Pro-Vxx-Aib-Aib/Vxx-Gln-Gln	[68]
Ⅱ	*T. parceramosum* CBS 936.69	Ac-Aib-Ala-Ala/Aib-Ala-Aib-Ala/Aib-Gln-Aib-Vxx-Aib-Gly-Lxx-Aib-Pro-Vxx-Aib-Aib/Vxx/Ala-Gln Ac-Aib-Ala-Ala/Aib-Ala-Aib-Ala/Aib-Gln-Aib-Vxx-Aib-Gly-Lxx-Aib-Pro-Vxx-Aib-Aib/Vxx/Ala-Gln-Gln-Pheol	[68]
III	*T. parceramosum* CBS 936.69	Ac-Aib-Asn/Gln-Vxx/Lxx-Vxx/Lxx-Aib-Pro-Lxx-Vxx/Lxx-Aib-Pro-Lxxol/Valol	[68]

Vxx: Val/Isovaline; Lxx: Leu/Ile; Lxxol: Leucinol/isoleucinol.

**Table 9 jof-06-00061-t009:** Analogs of trilongins

Trilongin	Strain	Composition	Ref.
AIV a	*T. longibrachiatum*	Ac-Aib-Asn-Vxx-Vxx-Aib-Pro-Vxx-Lxx-Aib-Pro-Lxxol	[77]
AIV b	*T. longibrachiatum*	Ac-Aib-Asn-Vxx-Vxx-Aib-Pro-Lxx-Lxx-Aib-Pro-Vxxol	[77]
AIV c	*T. longibrachiatum*	Ac-Aib-Asn-Vxx-Vxx-Aib-Pro-Lxx-Vxx-Aib-Pro-Lxxol	[77]
AIII a	*T. longibrachiatum*	Ac-Aib-Asn-Lxx-Vxx-Aib-Pro-Lxx-Lxx-Aib-Pro-Vxxol	[77]
AIII b	*T. longibrachiatum*	Ac-Aib-Asn-Lxx-Vxx-Aib-Pro-Vxx-Lxx-Aib-Pro-Lxxol	[77]
AIII c	*T. longibrachiatum*	Ac-Aib-Asn-Vxx-Lxx-Aib-Pro-Lxx-Lxx-Aib-Pro-Vxxol	[77]
AIII d	*T. longibrachiatum*	Ac-Aib-Asn-Vxx-Lxx-Aib-Pro-Vxx-Lxx-Aib-Pro-Lxxol	[77]
AII a	*T. longibrachiatum*	Ac-Aib-Asn-Lxx-Lxx-Aib-Pro-Lxx-Lxx-Aib-Pro-Vxxol	[77]
AII b	*T. longibrachiatum*	Ac-Aib-Asn-Lxx-Lxx-Aib-Pro-Lxx-Vxx-Aib-Pro-Lxxol	[77]
AII c	*T. longibrachiatum*	Ac-Aib-Asn-Lxx-Lxx-Aib-Pro-Vxx-Lxx-Aib-Pro-Lxxol	[77]
AII d	*T. longibrachiatum*	Ac-Aib-Asn-Lxx-Vxx-Aib-Pro-Lxx-Lxx-Aib-Pro-Lxxol	[77]
AII e	*T. longibrachiatum*	Ac-Aib-Asn-Vxx-Lxx-Aib-Pro-Lxx-Lxx-Aib-Pro-Lxxol	[77]
AI	*T. longibrachiatum*	Ac-Aib-Asn-Lxx-Lxx-Aib-Pro-Lxx-Lxx-Aib-Pro-Lxxol	[77]
A0	*T. longibrachiatum*	Ac-Aib-Gln-Lxx-Lxx-Aib-Pro-Lxx-Lxx-Aib-Pro-Lxxol	[77]
BI	*T. longibrachiatum*	Ac-Aib-Ala-Aib-Ala-Aib-Ala-Gln-Aib-Vxx-Aib-Gly-Lxx-Aib-Pro-Vxx-Aib-Aib-Gln-Gln-Pheol	[77]
BII	*T. longibrachiatum*	Ac-Aib-Ala-Aib-Ala-Aib-Ala-Gln-Aib-Vxx-Aib-Gly-Lxx-Aib-Pro-Vxx-Aib-Vxx-Gln-Gln-Pheol	[77]
BIII	*T. longibrachiatum*	Ac-Aib-Ala-Aib-Ala-Aib-Aib-Gln-Aib-Vxx-Aib-Gly-Lxx-Aib-Pro-Vxx-Aib-Aib-Gln-Gln-Pheol	[77]
BIV	*T. longibrachiatum*	Ac-Aib-Ala-Aib-Ala-Aib-Aib-Gln-Aib-Vxx-Aib-Gly-Lxx-Aib-Pro-Vxx-Aib-Vxx-Gln-Gln-Pheol	[77]
CI	*T. atroviride* H1/226	Ac-Aib-Ala-Aib-Ala-Aib-Ala-Gln-Aib-Vxx-Aib-Gly-Lxx-Aib-Pro-Vxx-Aib-Aib-Glu-Gln-Pheol	[76]
CII	*T. atroviride* H1/226	Ac-Aib-Ala-Aib-Ala-Aib-Ala-Gln-Aib-Vxx-Aib-Gly-Lxx-Aib-Pro-Vxx-Aib-Vxx-Glu-Gln-Pheol	[76]
CIII	*T. atroviride* H1/226	Ac-Aib-Ala-Aib-Ala-Aib-Aib-Gln-Aib-Vxx-Aib-Gly-Lxx-Aib-Pro-Vxx-Aib-Aib-Glu-Gln-Pheol	[76]
CIV	*T. atroviride* H1/226	Ac-Aib-Ala-Aib-Ala-Aib-Aib-Gln-Aib-Vxx-Aib-Gly-Lxx-Aib-Pro-Vxx-Aib-Vxx-Glu-Gln-Pheol	[76]

Lxx: Leu/Ile; Vxx: Val/Iva; Lxxol: Leuol/Ileol; Vxxol: Valol/Ivaol.

**Table 10 jof-06-00061-t010:** Analogs of trichokonins.

Trichokonin	Strain	Composition	Ref.
VI	*T. koningii* OUDEMANS, *T. pseudokoningii* SMF2	Ac-Aib-Ala-Aib-Ala-Aib-Ala-Gln-Aib-Val-Aib-Gly-Leu-Aib-Pro-Val-Aib-Aib-Gln-Gln-Pheol	[81,82]
IX	*T. koningii* OUDEMANS	Ac-Aib-Ala-Aib-Ala-Aib-Ala-Gln-Aib-Val-Aib-Gly-Leu-Aib-Pro-Val-Aib-Iva-Gin-Gln-Pheol	[81]
Ia	*T. koningii* OUDEMANS	Ac-Aib-Ala-Aib-Ala-Aib-Aib-Gln-Aib-Val-Aib-Gly-Leu-Ala- Pro-Val-Aib-Aib-Gln-Gln-Pheol	[81]
Ib	*T. koningii* OUDEMANS	Ac-Aib-Gly-Aib-Ala-Aib-Aib-Gln-Aib-Val-Aib-Gly-Leu-Aib-Pro-Val-Aib-Aib-Gln-Gln-Pheol	[81,84]

**Table 11 jof-06-00061-t011:** Analogs of alamethicins

Alamethicin	Composition
F30	Ac-Aib-Pro-Aib-Ala-Aib-Ala-Gln-Aib-Val-Aib-Gly-Leu-Aib-Pro-Val-Aib-Aib-Glu-Gln-Pheol
F50	Ac-Aib-Pro-Aib-Ala-Aib-Ala-Gln-Aib-Val-Aib-Gly-Leu-Aib-Pro-Val-Aib-Aib-Gln-Gln-Pheol

**Table 12 jof-06-00061-t012:** Analogs of trichotoxins.

Trichotoxin	Strain	Composition	Ref.
A-40	*T. viride* NRRL 5242	Ac-Aib-Gly-Aib-Leu-Aib-Glu-Aib-Aib-Aib-Ala-Aib-Aib-Pro-Leu-Aib-D-Iva-Gln-Valol	[90]
A-40 I	*T. viride* NRRL 5242	Ac-Aib-Gly-Aib-Leu-Aib-Gln-Aib-Aib-Ala-Ala-Aib-Aib-Pro-Leu-Aib-Aib-Glu-Valol	[90]
A-40 II	*T. viride* NRRL 5242	Ac-Aib-Gly-Aib-Leu-Aib-Gln-Aib-Aib-Aib-Ala-Ala-Aib-Pro-Leu-Aib-Aib-Glu-Valol	[90]
A-40 III	*T. viride* NRRL 5242	Ac-Aib-Gly-Aib-Leu-Aib-Gln-Aib-Aib-Ala-Ala-Aib-Aib-Pro-Leu-Aib-D-Iva-Glu-Valol	[90]
A-40 IV	*T. viride* NRRL 5242	Ac-Aib-Gly-Aib-Leu-Aib-Gln-Aib-Aib-Aib-Ala-Aib-Aib-Pro-Leu-Aib-Aib-Glu-Valol	[90]
A-40 V	*T. viride* NRRL 5242	Ac-Aib-Gly-Aib-Leu-Aib-Gln-Aib-Aib-Aib-Ala-Aib-Aib-Pro-Leu-Aib-D-Iva-Glu-Valol	[90]
A-40 Va	*T. viride* NRRL 5242	Ac-Aib-Ala-Aib-Leu-Aib-Gln-Aib-Aib-Aib-Ala-Aib-Aib-Pro-Leu-Aib-Aib-Glu-Valol	[90]
A-50 E	*T. viride* NRRL 5242	Ac-Aib-Gly-Aib-Leu-Aib-Gln-Aib-Aib-Aib-Ala-Ala-Aib-Pro-Leu-Aib-Aib-Gln-Valol	[90]
A-50 F	*T. viride* NRRL 5242	Ac-Aib-Gly-Aib-Leu-Aib-Gln-Aib-Aib-Ala-Ala-Ala-Aib-Pro-Leu-Aib-D-Iva-Gln-Valol	[90]
A-50 G	*T. viride* NRRL 5242	Ac-Aib-Gly-Aib-Leu-Aib-Gln-Aib-Aib-Aib-Ala-Ala-Aib-Pro-Leu-Aib-D-Iva-Gln-Valol	[90]
A-50 H	*T. viride* NRRL 5242	Ac-Aib-Ala-Aib-Leu-Aib-Gln-Aib-Aib-Aib-Ala-Ala-Aib-Pro-Leu-Aib-D-Iva-Gln-Valol	[90]
A-50 I	*T. viride* NRRL 5242	Ac-Aib-Gly-Aib-Leu-Aib-Gln-Aib-Aib-Aib-Ala-Aib-Aib-Pro-Leu-Aib-D-Iva-Gln-Valol	[90]
A-50 J	*T. viride* NRRL 5242	Ac-Aib-Ala-Aib-Leu-Aib-Gln-Aib-Aib-Aib-Ala-Aib-Aib-Pro-Leu-Aib-D-Iva-Gln-Valol	[90]

**Table 13 jof-06-00061-t013:** Analogs of suzukacillins.

Suzukacillin	Strain	Composition	Ref.
SZ-A1	*T. viride* 63C-I	Ac-Aib-Ala-Aib-Ala-Aib-Ala-Gln-Aib-Vxx-Aib-Gly-Aib-Aib-Pro-Vxx-Aib-Aib-Gln-Gln-Pheol	[94]
SZ-A2	*T. viride* 63C-I	Ac-Aib-Ala-Aib-Ala-Aib-Ala-Gln-Aib-Lxx-Aib-Gly-Aib-Aib-Pro-Vxx-Aib-Aib-Gln-Gln-Pheol	[94]
SZ-A3	*T. viride* 63C-I	Ac-Aib-Ala-Aib-Ala-Aib-Ala-Gln-Aib-Vxx-Aib-Gly-Aib-Aib-Pro-Vxx-Aib-Vxx-Gln-Gln-Pheol	[94]
SZ-A4	*T. viride* 63C-I	Ac-Aib-Ala-Aib-Ala-Aib-Ala-Gln-Aib-Lxx-Aib-Gly-Aib-Aib-Pro-Vxx-Aib-Vxx-Gln-Gln-Pheol	[94]
SZ-A5	*T. viride* 63C-I	Ac-Aib-Ala-Aib-Ala-Aib-Aib-Gln-Aib-Lxx-Aib-Gly-Aib-Aib-Pro-Vxx-Aib-Aib-Gln-Gln-Pheol	[94]
SZ-A6	*T. viride* 63C-I	Ac-Aib-Ala-Aib-Ala-Aib-Ala-Gln-Aib-Lxx-Aib-Gly-Aib-Aib-Pro-Vxx-Aib-Aib-Gln-Gln-Pheol	[94]
SZ-A7	*T. viride* 63C-I	Ac-Aib-Ala-Aib-Ala-Aib-Aib-Gln-Aib-Lxx-Aib-Gly-Aib-Aib-Pro-Vxx-Aib-Vxx-Gln-Gln-Pheol	[94]
SZ-A8	*T. viride* 63C-I	Ac-Aib-Ala-Aib-Ala-Aib-Ala-Gln-Aib-Lxx-Aib-Gly-Aib-Aib-Pro-Vxx-Aib-Vxx-Gln-Gln-Pheol	[94]
SZ-A9	*T. viride* 63C-I	Ac-Aib-Ala-Aib-Ala-Aib-Ala-Gln-Aib-Lxx-Aib-Gly-Lxx-Aib-Pro-Vxx-Aib-Aib-Gln-Gln-Pheol	[94]
SZ-A10a	*T. viride* 63C-I	Ac-Aib-Ala-Aib-Ala-Aib-Ala-Gln-Aib-Vxx-Aib-Gly-Lxx-Aib-Pro-Vxx-Aib-Vxx-Gln-Gln-Pheol	[94]
SZ-A10b	*T. viride* 63C-I	Ac-Aib-Ala-Aib-Ala-Aib-Ala-Gln-Aib-Lxx-Aib-Gly-Lxx-Aib-Pro-Vxx-Aib-Vxx-Gln-Gln-Pheol	[94]
SZ-A11a	*T. viride* 63C-I	Ac-Aib-Ala-Aib-Ala-Aib-Ala-Gln-Aib-Lxx-Aib-Gly-Lxx-Aib-Pro-Vxx-Aib-Aib-Gln-Gln-Pheol	[94]
SZ-A11b	*T. viride* 63C-I	Ac-Aib-Ala-Aib-Ala-Aib-Ala-Gln-Aib-Lxx-Aib-Gly-Lxx-Aib-Pro-Vxx-Aib-Vxx-Gln-Gln-Pheol	[94]
SZ-A12	*T. viride* 63C-I	Ac-Aib-Ala-Aib-Ala-Aib-Aib-Gln-Aib-Lxx-Aib-Gly-Lxx-Aib-Pro-Vxx-Aib-Vxx-Gln-Gln-Pheol	[94]
SZ-A13	*T. viride* 63C-I	Ac-Aib-Ala-Aib-Ala-Aib-Ala-Gln-Aib-Lxx-Aib-Gly-Lxx-Aib-Pro-Vxx-Aib-Vxx-Gln-Gln-Pheol	[94]
SZ-B1	*T. viride* 63C-I	Ac-Aib-Gln-Vxx-Lxx-Aib-Pro-Lxx-Lxx-Ala-Pro	[95]
SZ-B2	*T. viride* 63C-I	Ac-Aib-Gln-Lxx-Vxx-Ala-Pro-Lxx-Lxx-Aib-Pro-Vxxol	[95]
SZ-B3	*T. viride* 63C-I	Ac-Aib-Asn-Vxx-Vxx-Aib-Pro-Lxx-Lxx-Aib-Pro-Lxxol	[95]
SZ-B4	*T. viride* 63C-I	Ac-Ala-Gln-Vxx-Lxx-Aib-Pro-Lxx-Lxx-Aib-Pro-Lxxol	[95]
SZ-B5	*T. viride* 63C-I	Ac-Ala-Gln-Lxx-Lxx-Aib-Pro-Lxx-Lxx-Aib-Pro-Vxxol	[95]
SZ-B6	*T. viride* 63C-I	Ac-Ala-Gln-Lxx-Lxx-Aib-Pro-Lxx-Lxx-Aib-Pro-Lxxol -	[95]
SZ-B7	*T. viride* 63C-I	Ac-Ala-Gln-Lxx-Lxx-Aib-Pro-Lxx-Lxx-Aib-Pro-Vxxol	[95]
SZ-B8	*T. viride* 63C-I	Ac-Aib-Asn-Vxx-Lxx-Aib-Pro-Lxx-Lxx-Aib-Pro-Lxxol	[95]
SZ-B9	*T. viride* 63C-I	Ac-Aib-Gln-Lxx-Lxx-Aib-Pro-Lxx-Lxx-Aib-Pro-Vxxol	[95]
SZ-B10	*T. viride* 63C-I	Ac-Aib-Gln-Lxx-Vxx-Aib-Pro-Lxx-Lxx-Aib-Pro-Lxxol	[95]
SZ-B11	*T. viride* 63C-I	Ac-Aib-Asn-Lxx-Lxx-Aib-Pro-Lxx-Lxx-Aib-Pro-Lxxol	[95]
SZ-B12	*T. viride* 63C-I	Ac-Aib-Gln-Lxx-Lxx-Aib-Pro-Lxx-Lxx-Aib-Pro-Lxxol	[95]
SZ-B13	*T. viride* 63C-I	Ac-Vxx-Gln-Lxx-Lxx-Aib-Pro-Lxx-Lxx-Aib-Pro-Lxxol	[95]
SZ-B14	*T. viride* 63C-I	Ac-Aib-Ala-Lxx-Lxx-Aib-Pro-Lxx-Lxx-Aib-Pro-Lxxol	[95]
SZ-B15	*T. viride* 63C-I	Ac-Vxx-Glu-Lxx-Lxx-Aib-Pro-Lxx-Lxx-Aib-Pro-Lxxol	[95]

Vxx: Val or Iva; Lxx: Leu or Ile (isovaline); Vxxol: Valol or Ivaol; Lxxol: Leuol or Ileol.

**Table 14 jof-06-00061-t014:** Analogs of hypomurocins.

Hypomurocin	Strain	Composition	Ref.
A-1	*H. muroiana* IFO31288	Ac-Aib-Gln-Val-Val-Aib-Pro-Leu-Leu-Aib-Pro-Leuol	[96]
A-2	*H. muroiana* IFO31288	Ac-D-Iva-Gln-Val-Val-Aib-Pro-Leu-Leu-Aib-Pro-Leuol	[96]
A-3	*H. muroiana* IFO31288	Ac-Aib-Gln-Val-Leu-Aib-Pro-Leu-Ile-Aib-Pro-Leuol	[96]
A-4	*H. muroiana* IFO31288	Ac-Aib-Gln-Ile-Val-Aib-Pro-Leu-Leu-Aib-Pro-Leuol	[96]
A-5	*H. muroiana* IFO31288	Ac-Aib-Gln-Ile-Ile-Aib-Pro-Leu-Leu-Aib-Pro-Leuol	[96]
A-5a	*H. muroiana* IFO31288	Ac-Aib-Gln-Ile-Leu-Aib-Pro-Leu-Ile-Aib-Pro-Leuol	[96]
B-1	*H. muroiana* IFO31288	Ac-Aib-Ser-Ala-Leu-Aib-Gln-Aib-Val-Aib-Gly-Aib-Aib-Pro-Leu-Aib-Aib-Gln-Valol	[96]
B-2	*H. muroiana* IFO31288	Ac-Aib-Ser-Ala-Leu-Aib-Gln-Aib-Val-Aib-Gly-Aib-Aib-Pro-Leu-Aib-Aib-Gln-Leuol	[96]
B-3a	*H. muroiana* IFO31288	Ac-Aib-Ala-Ala-Leu-Aib-Gln-Aib-Val-Aib-Gly-Aib-Aib-Pro-Leu-Aib-Aib-Gln-Valol	[96]
B-3b	*H. muroiana* IFO31288	Ac-Aib-Ser-Ala-Leu-Aib-Gln-Iva-Val-Aib-Gly-Aib-Aib-Pro-Leu-Aib-Aib-Gln-Valol	[96]
B-4	*H. muroiana* IFO31288	Ac-Aib-Ser-Ala-Leu-Aib-Gln-Aib-Val-Aib-Gly-Iva-Aib-Pro-Leu-Aib-Aib-Gln-Valol	[96]
B-5	*H. muroiana* IFO31288	Ac-Aib-Ser-Ala-Leu-Aib-Gln-Aib-Val-Aib-Gly-Iva-Aib-Pro-Leu-Aib-Aib-Gln-Leuol	[96]
B	*T. harzianum*	Ac-Aib-Ser-Ala-Leu-Ala-Gln-Aib-Val-Aib-Gly-Aib-Aib-Pro-Leu-Aib- Aib-Gln-Valol	[59]

**Table 15 jof-06-00061-t015:** Analogs of atroviridins.

Atroviridin	Composition
A	Ac-Aib-Pro-Aib-Ala-Aib-Ala-Gln-Aib-Val-Aib-Gly-Leu-Aib-Pro-Val-Aib-Aib-Gln-Gln-Pheol
B	Ac-Aib-Pro-Aib-Ala-Aib-Ala-Gln-Aib-Val-Aib-Gly-Leu-Aib-Pro-Val-Aib-Iva-Gln-Gln-Pheol
C	Ac-Aib-Pro-Aib-Ala-Aib-Aib-Gln-Aib-Val-Aib-Gly-Leu-Aib-Pro-Val-Aib-Iva-Gln-Gln-Pheol

**Table 16 jof-06-00061-t016:** Analogs of neoatroviridins.

Neoatroviridin	Composition
A	Ac-Aib-Gly-Ala-Leu-Aib-Gln-Aib-Leu-Aib-Gly-Iva-Aib-Pro-Leu-Aib-Aib-Gln-Leuol
B	Ac-Aib-Gly-Ala-Leu-Iva-Gln-Aib-Leu-Aib-Gly-Iva-Aib-Pro-Leu-Aib-Aib-Gln-Leuol
C	Ac-Aib-Gly-Ala-Leu-Aib-Gln-Iva-Leu-Aib-Gly-Iva-Aib-Pro-Leu-Aib-Aib-Gln-Leuol
D	Ac-Aib-Gly-Ala-Leu-Iva-Gln-Iva-Leu-Aib-Gly-Iva-Aib-Pro-Leu-Aib-Aib-Gln-Leuol

**Table 17 jof-06-00061-t017:** Analogs of peptaivirins.

Peptaivirin	Composition
A	Ac-Phe-Aib-Ala-Aib-Iva-Leu-Gln-Gly-Aib-Aib-Ala-Ala-Aib-Pro-Iva-Aib-Aib-Gln-Trpol
B	Ac-Phe-Aib-Ser-Aib-Iva-Leu-Gln-Gly-Aib-Aib-Ala-Ala-Aib-Pro-Iva-Aib-Aib-Gln-Pheol

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
