# Peer review of "Diversity of Linear Non-Ribosomal Peptide in Biocontrol Fungi"

_jof, 2020, doi:10.3390/jof6020061_

Round 1

Reviewer 1 Report

The present manuscript (MS) is a review on the Diversity of Linear Non-Ribosomal Peptide in Biocontrol Fungi. The topic is interesting in view of the complexity of these metabolites, their abundance in fungi and their biological roles and potential applications. Authors however, need to pay a big effort for improving their MS. I have gone through the text and have corrected the text as much as possible to help them in their task of improving the document. However of a journal devoted to a broad audience such as JoF they need to improve not only the current text but to write a more attractive and comprehensive review on LNRPs. The text is in many parts too descriptive. There are tables which show slight chemical modifications of the structures which are of interest only to organic chemists specialized in synthetic organic chemistry. We instead lack conceptual diagrams on the structures of LNRPs with structure-function cartoons and the different families of compounds compared (it is imposible to do it in the MS now). The paper lacks fungal biology and evolutionary biology (use sequence data, phylogenetic trees...) related with secondary metabolism (which is what it is underneath the paper). A section of the function of the compounds for the two ecological groups of BF (entomopathogenic and mycoparasitic) is also lacking. The descriptions of the mode of action of some of the compounds specially on the membrane should be better exploited with attractive diagrams and the text changed accordingly. Regarding the style/grammar there are repetitive mistakes throughout the text (singulars for plurals and viceversa, etc..., ). The discussion lacks telling a story and is a list of data again. References are quite old. Nothing in the text makes you feel that we are in the DNA massive sequencing era (in the NGS actually), there are only some references to gene cloning which are from outdated papers. The references have lots of mistakes too (italics missing, strange symbols-computer codes-, no italics). There are lots of papers quoted written in chinese (would be the same with any other not english language), which is not the best for a review which aims a global audience. Please follow my comments on the pdf file and in this latter and make a more attractive manuscript.

Author Response

Dear reviewer,

We cordially thank you very much for your carfully reviewing our MS. We agree all your comments. Now, we revised the MS. We hope you give more correction and guidance.

  1. We accepted your revision of MS.
  2. We almost rewrote the MS in native English and do best to organize a better story, although it is difficult to us.
  3. We revised the all tables and figures for simpler and clearer.
  4. We rewrote the discussion making it a logic order.
  5. We corrected and updated the references, deleted many old documents and added new ones.

Best regards

Qiongbo HU

Reviewer 2 Report

This review summarizes what is known about non-ribosomal linear peptides in some fungi (entomopathogenic and mycoparasitic ones). The review appears sound but is poorly written with many grammatical and orthographic errors, as to make it difficult to understand in most parts. I have uploaded my corrections in the associated file, but I think that even more extensive editing is required. My only other remarks are:

(1) are linear peptides also known from other fungal groups, if so states which ones, even if comprehensive review is not given.

(2) some data could be condensed, especially all structures could be put together in a single figure and some tables could be combined together.

(3) abbreviations for scientific names does not follow the rules. 

Author Response

Dear reviewer,

We cordially thank you for carefully reviewing the MS. We agree all your comments. Please give more correct and guidance to our revision.

  1. We accepted the all your revision in MS.
  2. We almost re-wrote the MS with native English and composed the MS a better story as we hope.
  3. Of course, other fungal group produce linear peptides, but we fucos on the BFs L-NRPs.
  4. We revised all tables and figures to make them more simple and clear.
  5. We correct all the abbreviations for scientific names and updated the references. 

Best regards

Qiongbo HU

Round 2

Reviewer 1 Report

Dear Authors,

The MS is much improved from the previous version. I much like the discussion with the comparison of NRPs abundance in mycoparasites vs entomopathogens and the possible role of proteases of the latter in that. The New Fig. 10 will help the non expert reader to grasp the complexity of these secondary metabolites and in general the text has been edited extensively. Congratulations

Reviewer 2 Report

The paper is much improved. I nonetheless recommend a final English check before publication, as few errors remain.